# Prenatal Ultrasound Diagnosis of Klippel–Trenaunay Syndrome

**DOI:** 10.3390/diagnostics13223400

**Published:** 2023-11-08

**Authors:** Nicolae Gică, Andreea Dumitru, Anca Maria Panaitescu, Corina Gică, Gheorghe Peltecu, Anca Marina Ciobanu, Laura Bălănescu

**Affiliations:** 1Obstetrics and Gynecology Department, Faculty of Medicine, “Carol Davila” University of Medicine and Pharmacy, 020021 Bucharest, Romania; gica.nicolae@umfcd.ro (N.G.); mat.corina@gmail.com (C.G.); gheorghe.peltecu@umfcd.ro (G.P.); anca.ciobanu@umfcd.ro (A.M.C.); 2Department of Obstetrics and Gynecology, Filantropia Clinical Hospital Bucharest, 020021 Bucharest, Romania; andreea-elena.dumitru@rez.umfcd.ro; 3Department of Pediatric Surgery and Anaesthesia and Intensive Care, “Carol Davila” University of Medicine and Pharmacy, 020021 Bucharest, Romania; laura.balanescu@umfcd.ro

**Keywords:** prenatal diagnosis, vascular malformation, lymphatic abnormalities

## Abstract

Klippel–Trenaunay syndrome (KTS) is a very rare vascular malformation syndrome also referred to as a capillary–lymphatic–venous malformation with unknown aetiology. The aim of our paper is to highlight interesting images, regarding a rare case of foetal Klippel–Trenaunay syndrome diagnosed prenatally in our department and confirmed postnatally with a favourable evolution during the gestation and neonatal periods. This case was diagnosed at 26 weeks gestation and characterised through ultrasound by the presence of superficial multiple cystic structures of different sizes spreading over the left leg with hemihypertrophy and reduced mobility. The cystic lesions were spreading to the left buttock and the pelvic area. The right leg and upper limbs had normal appearance with good mobility. There were no signs of hyperdynamic circulation or foetal anaemia, but mild polyhydramnios was associated. The ultrasound findings were confirmed postnatally, the left leg presented multiple cystic lesions and port wine stains, and there was hypertrophy and fixed position, with favourable evolution at 6 months of life, when the size of the lesions began to decrease and the mobility of the leg improved.

**Figure 1 diagnostics-13-03400-f001:**
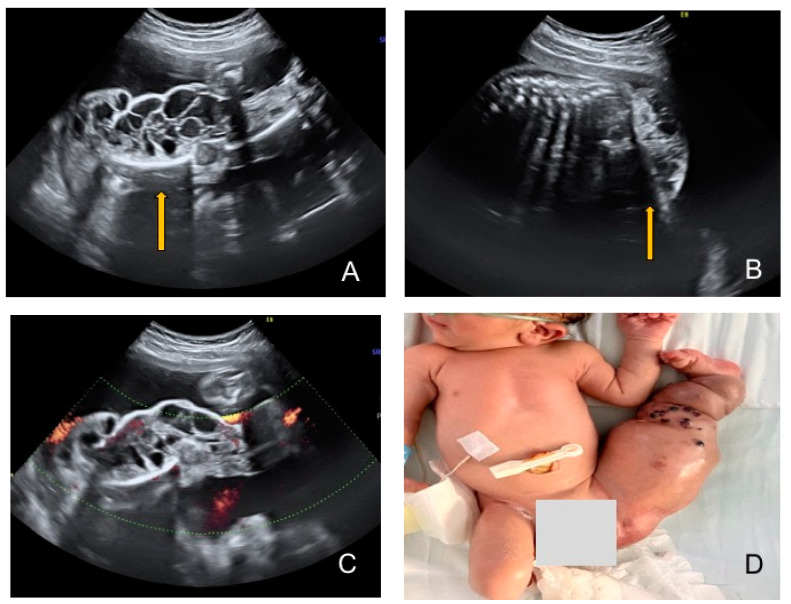
(**A**) G1 P1, who was referred to our unit at 26 weeks’ gestation for second opinion after the second-trimester morphological scan revealed a foetal vascular malformation affecting the lower limb. Ultrasound evaluation showed superficial multiple cystic structures of different sizes spreading over the left leg (arrow) with hemihypertrophy and reduced mobility. The thigh was in fixed flexed position on the abdomen and the calf in extension. (**B**) The sonographic interpretation in the second trimester revealed that cystic lesions were spreading to the left buttock and pelvic area (arrow). The right leg and upper limbs had normal appearance with good mobility. (**C**) A detailed ultrasound evaluation was performed in order to exclude involvement of other organs, and colour Doppler was helpful in assessing the presence and intensity of blood flow signals within the lesions. There were no signs of hyperdynamic circulation or foetal anaemia, but mild polyhydramnios was associated. Although the prenatal genetic testing was offered at the time of diagnosis through amniocentesis, the procedure was declined, but genetic counselling and WES (whole-exome sequencing) analysis was performed postnatally, which did not reveal any genetic mutation. After counselling, the couple continued the pregnancy without any other complications, the evolution was favourable, there was no extension of the lesions to the trunk or an increase in size with advancing gestation, and the delivery took place at 39 weeks. (**D**) A male newborn weighing 3200 g with Apgar score of 9 was delivered via caesarean section. The ultrasound findings were confirmed postnatally: the left leg presented multiple cystic lesions, skin lesions in the form of swollen blood vessels with reddish discolouration of the skin (port wine stains), hypertrophy and fixed position of the limbs. Klippel–Trenaunay syndrome is a clinical diagnosis characterised by the presence of localized cutaneous capillary malformations, venous abnormalities and bony or soft-tissue hypertrophy of an extremity [1,2]. Although most cases appear sporadically without the association of chromosomal abnormalities, the literature mentions reciprocal translocation, variants of familial occurrence with autosomal recessive inheritance or autosomal dominant types of inheritance with incomplete penetrance [3,4]. Many individuals with severe forms of KTS have been found to have *PIK3CA* (phosphatidylinositol-3 kinase) gene mutations, involved in making a protein that helps regulate cell growth, division and survival [5]. Prenatal diagnosis can be achieved usually as early as 20 weeks and the most common ultrasound findings include cystic lesions affecting mostly the limbs, either uni- or bilateral, but in almost 50% of cases the cystic lesions can affect the internal organs (especially the lungs, bowel and liver) together with hemihypertrophy of the affected limb, anomalies of the fingers and visceromegaly [6,7]. Polyhydramnios, cardiomegaly, ascites, pleural effusion and hydrops that could be absent at the moment of the diagnosis appear frequently later in pregnancy. The intrauterine evolution might be variable, and the progression of the haemangiomas to the trunk and hypertrophy of the affected limb usually occur with advancing gestational age. The neonatal prognosis is generally favourable, but the quality of life might be severely affected; therefore, early prenatal diagnosis is an important criterion for decision making, allowing parents to decide whether to continue the pregnancy or not and for clinicians to carefully decide the time and place of delivery and to offer appropriate neonatal care. Severe complications such as consumption coagulopathy, cardiac failure, thrombosis, pulmonary embolism and trophic ulcers are the main causes of death. Kasabach–Merritt syndrome (disseminated intravascular coagulation and haemolysis) can complicate around 30% of cases [6,8]. This syndrome was included in the PIK3CA-related overgrowth disorders (PROSs) together with macrocephaly, vascular malformations of lower limbs, malformations of the head and neck, skin lesions and partial or generalised enlargement of one or more body part [9,10,11]. Differential diagnosis should be made with lymphangioma, hereditary lymphoedema and Parkes Weber syndrome (characterised by arteriovenous fistula, which can be identified using colour Doppler evaluation). Careful and repeated ultrasound monitoring is needed to evaluate foetal condition and to determine the time and the mode of delivery, depending on the gestational age, extent of the lesions and internal organ involvement. In cases with stable evolution, the treatment can be conservative and surgical. Neonatal interventions like sclerotherapy, laser therapy or radiotherapy can be performed, while in severe cases, where extensive lesions determine bleeding, infection or ulcers, limb amputation might be necessary. Klippel–Trenaunay syndrome is a complex condition diagnosed prenatally based on ultrasound cystic changes or postnatally based on characteristic signs and symptoms. Genetic testing might be offered but, in most cases, the karyotype is normal. Prenatal diagnosis with or without Doppler signal reveals multiple cystic lesions, which can increase in size and extent in different body areas. Extensive lesions might have a poor prognosis with a high mortality rate due to thrombosis or cardiac failure. There is no cure for KTS, and the therapeutic options are limited to different neonatal interventions, both surgical and non- surgical, aiming to improve the function and the aesthetic appearance of the affected area.

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
