# Peer review of "Prenatal Ultrasound Diagnosis of Klippel–Trenaunay Syndrome"

_diagnostics, 2023, doi:10.3390/diagnostics13223400_

Round 1
Reviewer 1 Report
Comments and Suggestions for Authors
Esteemed authors and editorial team,
This is a very interesting and rare prenatal diagnosis. I think the manuscript should be accepted provided some minor changes are made.
Abstract: It is not clear from the abstract this is the case of a prenatal ultrasound diagnosis of a fetus with KT syndrome confirmed postnatally. Please review.
Introduction – I would rather call it maintext, perhaps, since this is the only section of the manuscript
- Line 32 “anomalies of the digits” – should be changed to “anomalies of the fingers”. Digits means numbers.
- Line 36 “occur with the gestational age” – change to “occur with advancing gestational age”.
- by the end of the section therapeutic means are discussed, but it should be specified these are neonatal interventions. (lines 48-51)
Figure legend
- Line 58 “the shin in extension” – what does shin mean?
- I suggest reassigning the text to the figure letters. I would place the letter designating the image before the explanation. For example some of the text before letter D, clearly belongs to C, whose main description is before.
- Line 68 “port wine stains” – meaning???
I suggest a conclusive phrase.
Was karyotyping offered/provided at any time perinatally?
Comments on the Quality of English Language
I have included comments in my review. Review by a native speaker could be achieved to refine some phrases.
Author Response
Comments and Suggestions for Authors
Esteemed authors and editorial team,
This is a very interesting and rare prenatal diagnosis. I think the manuscript should be accepted provided some minor changes are made.
Response: Thank you.
Abstract: It is not clear from the abstract this is the case of a prenatal ultrasound diagnosis of a fetus with KT syndrome confirmed postnatally. Please review.
Response: Thank you, we have made this change in the abstract lines 13-15
Introduction – I would rather call it maintext, perhaps, since this is the only section of the manuscript
Response: Thank you, we have removed the introduction
- Line 32 “anomalies of the digits” – should be changed to “anomalies of the fingers”. Digits means numbers.
Response: Thank you, we have made the change
- Line 36 “occur with the gestational age” – change to “occur with advancing gestational age”.
Response: Thank you, we have made the change
- by the end of the section therapeutic means are discussed, but it should be specified these are neonatal interventions. (lines 48-51)
Response: Thank you, we have made the change line 59 and in the last paragraph
Figure legend
- Line 58 “the shin in extension” – what does shin mean?
Response: Thank you, we have made the change line 68
- I suggest reassigning the text to the figure letters. I would place the letter designating the image before the explanation. For example some of the text before letter D, clearly belongs to C, whose main description is before.
- Line 68 “port wine stains” – meaning???
Response: Thank you, we have made the change line 84-85
I suggest a conclusive phrase.
Response: Thank you, we have introduced the last paragraph
Was karyotyping offered/provided at any time perinatally?
Response: Thank you, we have added the comment in line 75-77
Comments on the Quality of English Language
I have included comments in my review. Review by a native speaker could be achieved to refine some phrases.
Response: Thank you, the revision was made.
Reviewer 2 Report
Comments and Suggestions for Authors
Considering that most of the more severe cases of Klippel-Trenaunay syndrome have been found to be caused by mosaic activating variants in the PIK3CA gene, please include related references.
Please proceed to English language corrections before publication.
Examples
Line 23
“Klippel-Trenaunay…” to “Klippel-Trenaunay syndrome…”
Line 38
“…an important criteria…” to “…an important criterion…”. Criteria is the plural form of criterion and should not be used as a singular noun.
Comments on the Quality of English LanguageEnglish language corrections are necessary before publication.
Author Response
Comments and Suggestions for Authors
Considering that most of the more severe cases of Klippel-Trenaunay syndrome have been found to be caused by mosaic activating variants in the PIK3CA gene, please include related references.
Response: Thank you, we have introduced the comment regarding PROS in line31-33 and 50-53 with the related references
Please proceed to English language corrections before publication.
Examples
Line 23
“Klippel-Trenaunay…” to “Klippel-Trenaunay syndrome…”
Line 38
“…an important criteria…” to “…an important criterion…”. Criteria is the plural form of criterion and should not be used as a singular noun.
Response: Thank you, we have made the changes
Reviewer 3 Report
Comments and Suggestions for Authors
Although the article provides a more typical ultrasound clinical phenotype, the content is simple and has little clinical significance. It provides practical diagnosis and treatment suggestions and tertiary prevention measures for clinical work to deal with the disease.
Comments on the Quality of English LanguageAverage
Author Response
Thank you for your comment
Klippel-Trenaunay syndrome is a very rare condition and most of the cases reported in the literature refer to the postnatal diagnosis, our report is highlighting the prenatal ultrasound diagnosis which is important for fetal medicine specialists dealing with such cases. All the case series reported prenatally ended in termination of pregnancy and about 35% of the ones continuing the pregnancy had a neonatal demise due to coagulopathy in the first hours/days of life. Our case also present a favourable evolution without complications neither during pregnancy, nor in the neonatal period. Although in the literature the prognosis of the disease is reported as being unfavourable with poor quality of life and large number of complications, our case can contribute to the counseling of parents, after an accurate diagnosis is made, in deciding to continue the pregnancy or not.
Round 2
Reviewer 2 Report
Comments and Suggestions for Authors
Minor editing of English language required
Comments on the Quality of English LanguageMinor editing of English language required